# Dosage Compensation of the X Chromosome during Sheep Testis Development Revealed by Single-Cell RNA Sequencing

**DOI:** 10.3390/ani12172169

**Published:** 2022-08-24

**Authors:** Jie Su, Yue Zhang, Hong Su, Caiyun Wang, Daqing Wang, Yanyan Yang, Xiunan Li, Wangmei Qi, Haijun Li, Xihe Li, Yongli Song, Guifang Cao

**Affiliations:** 1Inner Mongolia Key Laboratory of Basic Veterinary Science, Inner Mongolia Agriculture University, Hohhot 010018, China; 2Department of Psychosomatic Medicine, Inner Mongolia Medical University, Hohhot 010030, China; 3Inner Mongolia Academy of Agriculture & Animal Husbandry Sciences, Hohhot 010000, China; 4Inner Mongolia Saikexing Institutes of Breeding and Reproductive Biotechnologies in Domestic Animal, Hohhot 011517, China; 5Research Center for Animal Genetic Resources of Mongolia Plateau, College of Life Science, Inner Mongolia University, Hohhot 010021, China

**Keywords:** dosage compensation, single-cell RNA sequencing, sheep testis

## Abstract

**Simple Summary:**

Male and female mammals carry the same complement of autosomes but differ with respect to their sex chromosomes: females carry XX chromosomes and males carry XY chromosomes. The evolutionary loss of genes from the Y chromosome led to a disparity in the dosage of X chromosomes versus autosomal genes, with males becoming monosomic for X-linked gene products. An imbalance in gene expression may have detrimental consequences. In males, X-linked genes need to be upregulated to levels equal to those of females, which is called dosage compensation. The existence of dosage compensation in germ cells is controversial. In testis, dosage compensation is thought to cease during meiosis. Some studies showed that the X chromosome is inactivated during meiosis and premature transcriptional inactivation of the X and Y chromosome during mid-spermatogenesis is essential for fertility. However, some studies failed to find support for male germline X inactivation. Using single-cell RNA seq data, in this study, we presented a comprehensive transcriptional map of sheep testes at different developmental stages and found that germ cell types in sheep testes show X-chromosome expression similar to that in the autosomes. The dosage compensation of germ cells at different stages was analyzed. MSL complex members are expressed in female flies and orthologs exist in many species, where dosage compensation mechanisms are absent or fundamentally different. This suggests that the MSL complex members also function outside of the dosage compensation machinery. Studies have shown that MSL complex can regulate mammalian X inactivation and activation.

**Abstract:**

Dosage compensation is a mechanism first proposed by Susumu Ohno, whereby X inactivation balances X gene output between males (XY) and females (XX), while X upregulation balances X genes with autosomal gene output. These mechanisms have been actively studied in Drosophila and mice, but research regarding them lags behind in domestic species. It is unclear how the X chromosome is regulated in the sheep male germline. To address this, using single-cell RNA sequencing, we analyzed testes in three important developmental stages of sheep. We observed that the total RNA per cell from X and autosomes peaked in SSCs and spermatogonia and was then reduced in early spermatocytes. Furthermore, we counted the detected reads per gene in each cell type for X and autosomes. In cells experiencing dose compensation, close proximity to MSL (male-specific lethal), which is regulated the active X chromosome and was observed. Our results suggest that there is no dose compensation in the pre-meiotic germ cells of sheep testes and, in addition, MSL1 and MSL2 are expressed in early germ cells and involved in regulating mammalian X-chromosome inactivation and activation.

## 1. Introduction

In mammals, sex is determined by the heteromorphic sex chromosomes: XY chromosomes in males and XX chromosomes in females. Different genomic doses of sex chromosomes lead to unbalanced gene expression. The somatic expression of X-linked genes must be adjusted so that males and females produce similar levels of most proteins encoded on the variable chromosome [1]. Susumo Ohno suggested that to rectify this imbalance, the expression of X-chromosome genes should be increased two-fold to match the output of the diploid autosomal complement, i.e., to enable an X-to-autosome ratio (X:A) of 1 [2]. This is known as ‘dosage compensation’. Dosage compensation is achieved by increasing the X-chromosome expression levels of dosage-sensitive X-linked genes in both sexes and through the random silencing of one X chromosome in females (X-chromosome inactivation, XCl) [3,4].

Dosage compensation in animal somatic tissues has been extensively studied [4,5,6]. Two previous microarray studies fully supported dosage compensation in both humans and mice [5,7]. However, an RNA sequencing (RNA-seq) study in humans and mice claimed that microarray-based expression was not suitable for the comparison of the expression levels of different genes and there was no evidence of dosage compensation. Subsequently, the same RNA-seq data were reanalyzed in humans and mice and dosage compensation was found after taking into account the skewed gene content of the X chromosome [8,9]. X-chromosome upregulation in sheep has been supported by results from RNA sequencing (RNA seq) [4]. During male meiosis in some animal species, the X and Y become transcriptionally silenced in a process known as meiotic sex chromosome inactivation (MSCI) [10]. Following the initiation of MSCl, chromosomes undergo various genetic modification. However, whether dosage compensation occurs in germ cells is actively debated [11]. For instance, research has revealed the two-fold upregulation of X-linked genes in the germline of adult Drosophila testes [11,12]. Other research has found that the expression of the X chromosome is greatly suppressed in the male germline [13]. Further studies have suggested there is little or no evidence for dosage compensation and X-chromosome-specific inactivation during meiosis [14].

Despite the fact that dosage compensation in mammalian cells is fundamentally different compared to Drosophila [15], at least the core MSL complex consisting of MSL1, 2, 3 and MOF is conserved in mammalian species. The MSL complex in humans consists of hMSL1, hMSL2, hMSL3 and hMOF (hMYST1), whereas in mice, Hampin (Homolog of drosophila MSL1), MSL2, MRG15 and MYST1 were detected [16,17,18,19,20,21]. Remarkably, there is also evidence for a function of MOF and/or the MSL complex in upregulating the active X chromosome [8,22]. The study of the expression of the MSL complex during sheep spermatogenesis will further be essential for clarifying the role of the MSL complex in regulating mammalian X inactivation as well as activation.

The mechanism of germ cell and somatic cell dosage compensation in sheep testis remains unknown. In this study, we used RNA-seq data to study germ cell and somatic cell dosage compensation in sheep testes in different developmental stages. Meanwhile, we perform single-cell RNA-seq analysis to examine the timing of X:A balancing in newborn, juvenile and adult sheep testes.

## 2. Materials and Methods

### 2.1. Animal Care

The Institutional Animal Care and Use Committee of Inner Mongolia Agricultural University approved the experimental protocol employed in this study.

### 2.2. Reagents

All the reagents were purchased from Sigma–Aldrich Co. (St. Louis, MO, USA), unless otherwise indicated.

### 2.3. Animals and Sample Collection

Three healthy male Hu sheep at the ages of 0 days, 90 days and 720 days were used in this study. Hu sheep fed at the Aowoteke Animal Husbandry Technology Co., Ltd., Chifeng, China, were selected and castrated for testis collection. After the castrations were performed, the testes were removed and washed several times using DPBS, immediately transferred to Tissue Storage Solution (130-100-008, Miltenyi, Miltenyi Biotec B.V & CO, Mönchengladbach, Germany) at 4 °C and digested via two-step enzyme digestion within 8 h [8].

### 2.4. Preparation of Single Testicular Cell Suspension

The testis tissues were washed four times with DPBS and cut into small pieces. Small pieces of testicular tissue were washed four times with DPBS and placed in a 1.5 mL centrifuge tube. Single testicular cells were isolated using enzymatic digestion. In brief, the tissues were incubated in dissociation buffer containing collagenase IV 10 mg/mL, Dispase II 250 ug/mL, DNase I 1:1000 and 10% FBS in Modified Eagle’s Medium (DMEM, Gibco Laboratories, Grand City, NY, USA) for 30 min at 37 °C. The cell resuspensions were washed and resuspended and then filtered with through 40 μm strainers. The cell viability was determined via trypan blue staining and the cell numbers were determined using a hemocytometer (Countstar, New York, NY, USA). A cell suspension at a concentration of ~1000 cells/µL and cell viability >80% was used for single-cell RNA sequencing.

### 2.5. Single-Cell RNA-Sequencing Performance, Library Preparation and Sequencing

Single-cell RNA sequencing was performed using the 10× Genomics system. The single cells were loaded on a Chromium Single Cell Controller (10× Genomics, Pleasanton, CA, USA) to generate single-cell gel beads in emulsion (GEMs) by using the Single Cell 30 Library and Gel Bead Kit V2 (10× Genomics, 120237). The lysis and barcoded reverse transcription of polyadenylated mRNA from single cells were performed inside each GEM and they were cleaned. cDNA was amplified, then fragmented and fragment end repaired. An Agilent 2100 bioanalyzer (Agilent Technologies Inc., Santa Clara, CA, USA) and quantitative real-time PCR (qRT-PCR) were used to quantify the library. Finally, the qualified cDNA libraries were sequenced using the Illumina Hiseq 2000 (Illumina, Inc., Hayward, CA, USA).

### 2.6. Mapping, Cell Identification and Clustering Analysis

After low-quality reads were removed (nFeature_RNA < 200 or nFeature_RNA > 6000), Cell Ranger software (Cell Ranger v6.1.1, 10× Genomics, Pleasanton, CA, USA) was used to carry out gene expression quantification and cell-type identification based on the cell barcode and UMI information of read 1 in every single cell. The corresponding read 2 was mapped to the reference genome (Ovis aries, Oar_rambouillet_v1.0) using STAR software to determine the source genes of reads that completed the gene expression quantification. We used Seurat software and the PCA to classify the cell types. The filtering criteria of effective cells, including the number of genes per cell and the number of reads per cell, were kept in a range of 1–99% and the mitochondrial percentage was less than 5%. For the top 10 principal components with the largest variance explained in the PCA results, UMAP (uniform manifold approximation and projection) was used to visualize the single-cell clustering. The Louvain–Jaccard clustering method with a resolution of 0.4 was used to cluster and classify the cell types. Single-cell RNA-sequencing data were submitted to SRA repository through the access code: PRJNA865464.

### 2.7. Calculating Relative RNA Content from Each Cell Type

The expression level of each gene was calculated as the mean of the three biological replicates performed for each tissue in units of fragments per kilobase of transcript per million mapped reads (FPKM). Only genes with FPKM > 1 were included in our analysis [23]. After removing testis-specific or testis-biased genes and genes with effective lengths less than 100 bp, in total, 433 X genes and 13,003 autosome genes were present in the sheep genome. The relative RNA content per cell was estimated using the total RNA expression of X- and autosome-related genes in each cell divided by the number of cells. As each mRNA was randomly linked to one Unique Molecular Index (UMI/counts) after reverse transcription, the number of mRNA could be calculated by counting different UMIs. This is a proxy for the relative RNA content per cell, rather than a measurement of the actual number of RNA molecules present.

### 2.8. Comparing RNA Output by Chromosome and Spermatogenic Stage

In this study, we also detected the RNA content produced by individual X chromosome and autosomal genes. The total RNA counts from every X and autosomal gene in every cell type were calculated using sum of UMIs and by dividing this by nGene (the total number of genes on the X chromosomes/autosomes). We then log transformed these counts with y = Log2(counts + 1) and performed nondirectional Wilcoxon tests, with Holm-corrected *p* values indicating whether genes from the X chromosome were likely to have equal median counts to genes from the autosomes [12].

### 2.9. Immunofluorescence

Rabbit anti-MSL1 antibody (1:400 dilution, BS-178556R. Bioss, Beijing, China) was used for primary antibody incubation. Alexa Fluor 555-conjugated goat anti-rabbit IgG H&L (1:400) (BS-0295G-AF555. Bioss, Beijing, China) was used to produce immunofluorescence. Finally, DAPI was used for staining. Images were obtained using a confocal laser microscope (LSM, Zeiss, Oberkochen, Germany).

## 3. Results

### 3.1. Mapping, Cell Identification and Clustering Analysis

We obtained sheep testis tissues from newborn (0 days), juvenile (90 days) and adult (720 days) sheep. We sequenced 12,465 newborn, 12,388 juvenile and 8900 adult cells, and 11,000, 9811 and 5785 cells, respectively, were filtered for further analysis based on the filtering criteria. In total, 82.4%, 71.3% and 85.2% reads, respectively, were mapped to the sheep genome. In addition, 16,849, 17,571 and 17,441 genes were detected, and approximately 1314, 1151 and 1124 genes, respectively, were identified, on average, in each cell.

After sequencing, cell partitioning took place through the analysis of single-cell transcriptomes. We selected the marker genes of each testicular cell type to identify the cell type of each cluster [24,25,26,27,28]. Clustered corresponding cell types were obtained (Figure 1A). Newborn-stage testis cells were divided into 10 clusters, including 8 somatic cell clusters (macrophages, T cells, VSMs, endothelial, Sertoli cells, mesenchymal cells, Leydig cells and immature Leydig cells) and 2 germ cell clusters (SSCs and spermatogonia). In juvenile and adult sheep, the testis cells were divided into 15 clusters, including 8 somatic cell clusters and 7 germ cell clusters (SSCs, spermatogonia, early primary S’cytes, late primary S’cytes, spermatid-1, spermatids-2 and sperm) (Figure 1B,C and Appendix A).

### 3.2. X and Autosomal Total Transcription

To investigate the transcription regulation of X chromosomes and autosomes in sheep testes, we counted the Unique Molecular Indices (UMIs, a proxy for RNA content) from X chromosomes and autosomes in all the cells. We observed that the overall patterns for X and autosomes looked very similar in newborns, juveniles and adults (Figure 2 and Table 1), the ratios of X/autosome were the highest in somatic cells, intermediate in SSCs and spermatogonia cells and the lowest in meiotic (early primary S’cytes and late primary S’cytes) and post-meiotic (spermatids and sperm) cells. The relatively reduced ratio of X-chromosome RNA from spermatocytes to sperm suggested that the X chromosome was downregulated in excess of autosomes in meiotic and post-meiotic cells. It was observed that this X/autosome ratio was lower than that in somatic cells. If a cell type was dosage compensated, the ratio of X/A (X/autosome) would be similar to or higher than that found in somatic cells. If a cell type lacked dosage compensation, the ratio would be lower than that of somatic cells.

### 3.3. X Chromosome and Autosomal Relative RNA Production

As the total RNA content in cells cannot fully reflect the gene expression trend, we also detected the RNA content produced by individual genes of X chromosomes and autosomals at different stages of sperm development. We counted the detected reads per gene in each cell type for X chromosomes and autosomes. We carried out an analysis at different developmental stages and compared the median reads per gene between the X chromosomes and autosomes as a measure of X-to-autosome dosage compensation. The results showed that the median reads per gene from X genes were lower than those from autosomal genes; from this, we concluded that cell types are unaffected by dose compensation (Figure 3A–C and Table 2, adjusted *p* values < 0.05).

### 3.4. MSL Expression in Germ Cells

The core MSL complex consisting of MSL1, 2, 3 and MOF is conserved in mammalian species. There is no related study on sheep MSL complex. In our data, we found that only MSL1 and MSL2 were expressed in early germ cells, including SSCs, spermatogonia and early primary S’cytes (Figure 4A). The immunofluorescence staining results showed that MSL1 was located in the spermatogonia and SSCs of newborn stage; meanwhile, it was expressed in the SSCs, spermatogonia, early primary S’cyte and late primary S’cyte of the juvenile stage and adult stage (Figure 4B).

## 4. Discussion

In this study, we selected three important periods of sheep testes development. The two most important stages of testicular development are pre-maturity and post-maturity [29]. The newborn and juvenile period belongs to pre-maturity; in the former, seminiferous tubules contain infantile spermatogonia and undifferentiated Sertoli cells, and in the latter, both types of cells increase in number and spermatocytes and sperm appears gradually [30,31]. In adults, after maturity, spermatogonia immediately continue the proliferation and differentiation effect, representing an acceleration in the germ cell developmental process and establishing the functional seminiferous epithelium [32]. Firstly, we established an approach to successfully identify all types of testis cells from sheep at different stages with single-cell RNA seq.

In this work, we directly observed dosage compensation in terms of X:autosome count ratios via single-cell RNA seq. The results showed that the X:A ratio in germ cells was lower than that in somatic cells and the total RNA per cell from the X and autosomes reached its peak in SSCs and spermatogonia. Analyses showed a decreased X expression in early primary S’cytes, indicating that the sperm that undergoes meiotic sex chromosome inactivation (MSCI) brings in inactive X chromosomes. In mammals, the X chromosomes are inactivated during meiosis, possibly as a defense mechanism against insertions into unpaired chromatin [33]. This result is consistent with recent findings regarding Drosophila germ cells [12]. Due to the limitations of single-cell RNA seq, this study did not take into account any individual variation in gene expression at different developmental stages.

Somatic dosage compensation mechanisms in major genetic model organisms clearly do not function in the germline. Rastelli and Kuroda showed that MLE does not localize to X chromosomes and MSL3 is not expressed in the Drosophila germline cells of adult testes [34]. However, Gupta and colleagues assayed X-chromosome transcripts using DNA microarrays and revealed the two-fold upregulation of X-linked genes in the germline of adult Drosophila testes [7]. Recently, some transcriptome studies have shown that dosage compensation occurs in pre-meiotic germ cells, while others have shown that no dosage compensation occurs in the germline of adult Drosophila testes [11,13,14].

By comparing the RNA output, we found evidence that X chromosomes produced lower numbers of reads per gene than autosomes in germ cells. This finding is contrary to Witt‘s [12]. Our results showed that there was no dose compensation in the germ cells of sheep testis at different developmental stages. This is the first study of X-chromosome dosage compensation in sheep germ cells.

The MSL complex in male flies mediates the global acetylation of histone H4 lysine 16 (H4K16ac) on single male X chromosomes, leading to upregulation [35,36]. Remarkably, there is also evidence that clarifies the role of the MSL complex in regulating mammalian X inactivation as well as activation [37]. Studies have shown that MSLs do not function in the germline [34]. Notably, apart from MSL2, other MSL-complex-member orthologs exist in many species, in which dosage compensation mechanisms are absent or fundamentally different [38]. At present, there is no relevant study on MSL in sheep testis. In our study, only MSL1 and MSL2 were found. MSLs were located in premeiotic germ cells, which may be involved in regulating mammalian X-chromosome inactivation and activation.

## 5. Conclusions

Although many questions remain, our study strongly supports the absence of dose compensation in sheep pre-meiotic germ cells. MSL is not related to dose compensation but may be involved in regulating mammalian X-chromosome inactivation and activation.

## Figures and Tables

**Figure 1 animals-12-02169-f001:**
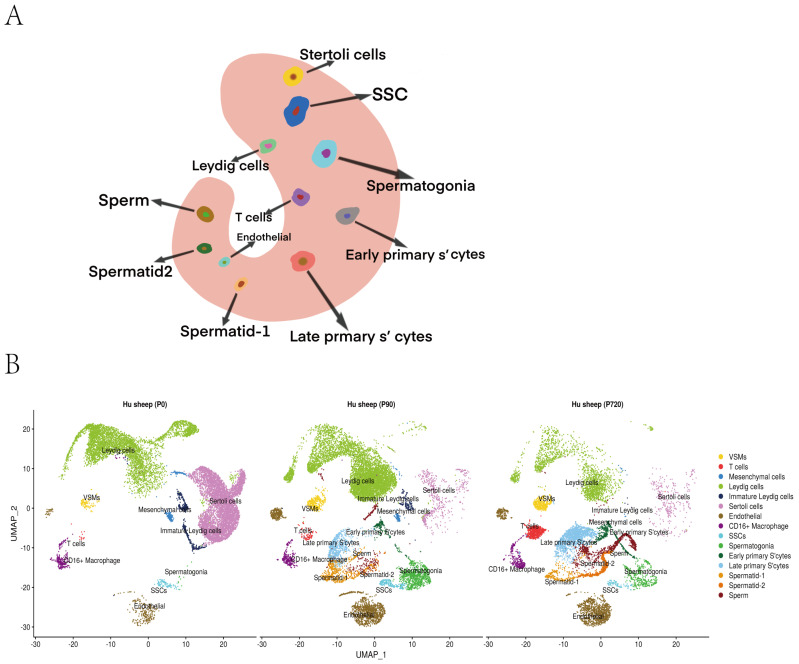
Clustering and cell−type assignment of single cells in Seurat. (**A**) An illustration of the major cell types in the testis. (**B**). A UMAP projection of every cell type identified in the data. (**C**). Expression distribution of marker genes in each cell types.

**Figure 2 animals-12-02169-f002:**
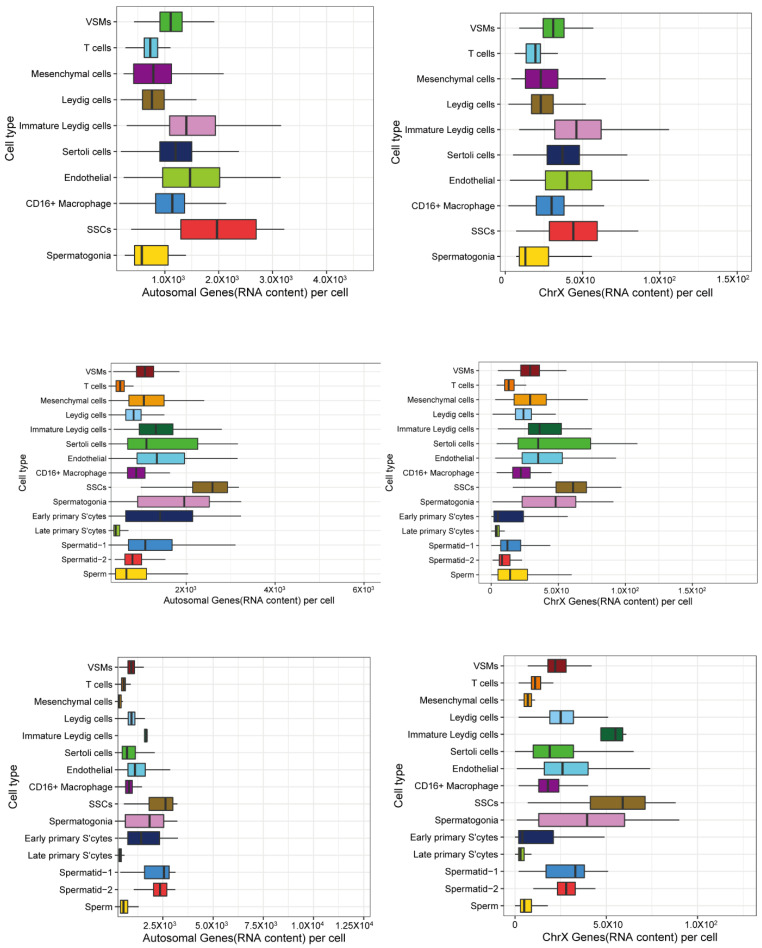
P0, P90, P720 relative RNA content from each cell type in testis. For every cell, the number of RNA molecules detected from the X chromosome and autosomes approximated by the summed number of Unique Molecular Indices counted per cell. Relative RNA content from each cell type in newborn (P0). Relative RNA content from each cell type in juvenile (P90). Relative RNA content from each cell type in adult. In juvenile and adult (P720).

**Figure 3 animals-12-02169-f003:**
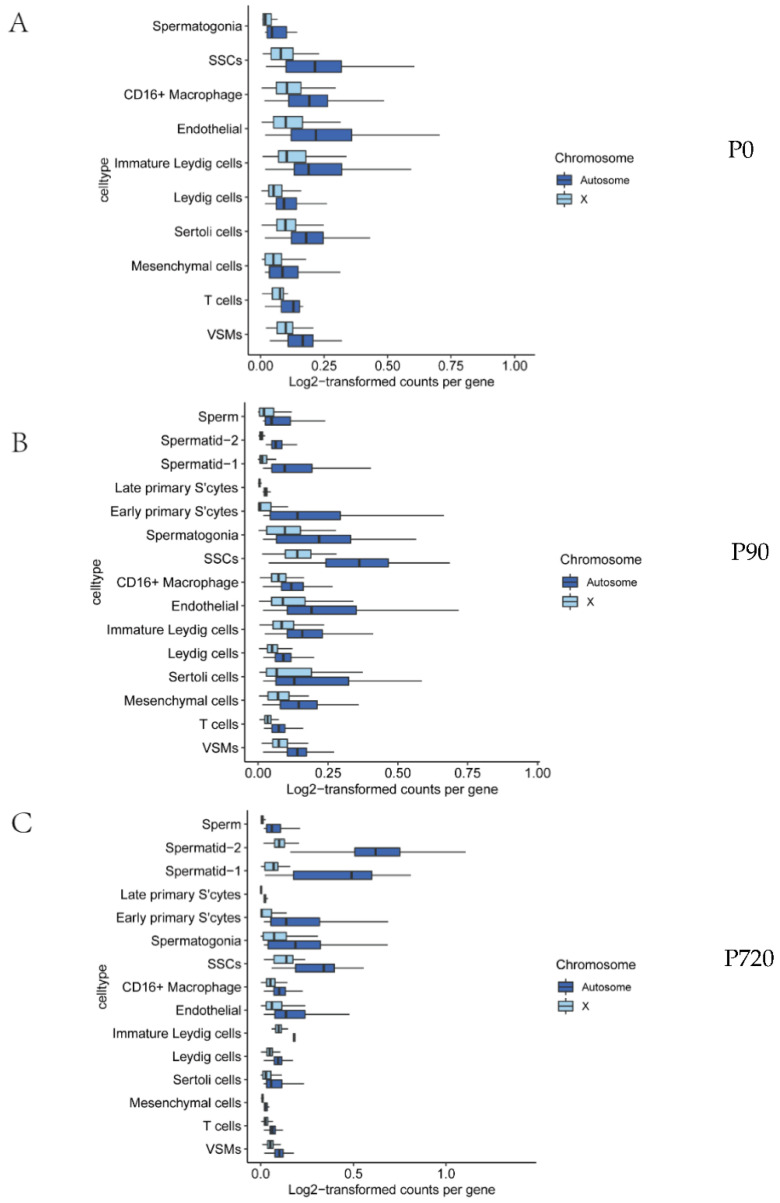
Dosage compensation equalizes X and autosomal transcription in select cell types. (**A**–
**C**) are the median log-transformed (log2(counts+1) counts for every gene in X and autosomes of P0, P90 and P720.

**Figure 4 animals-12-02169-f004:**
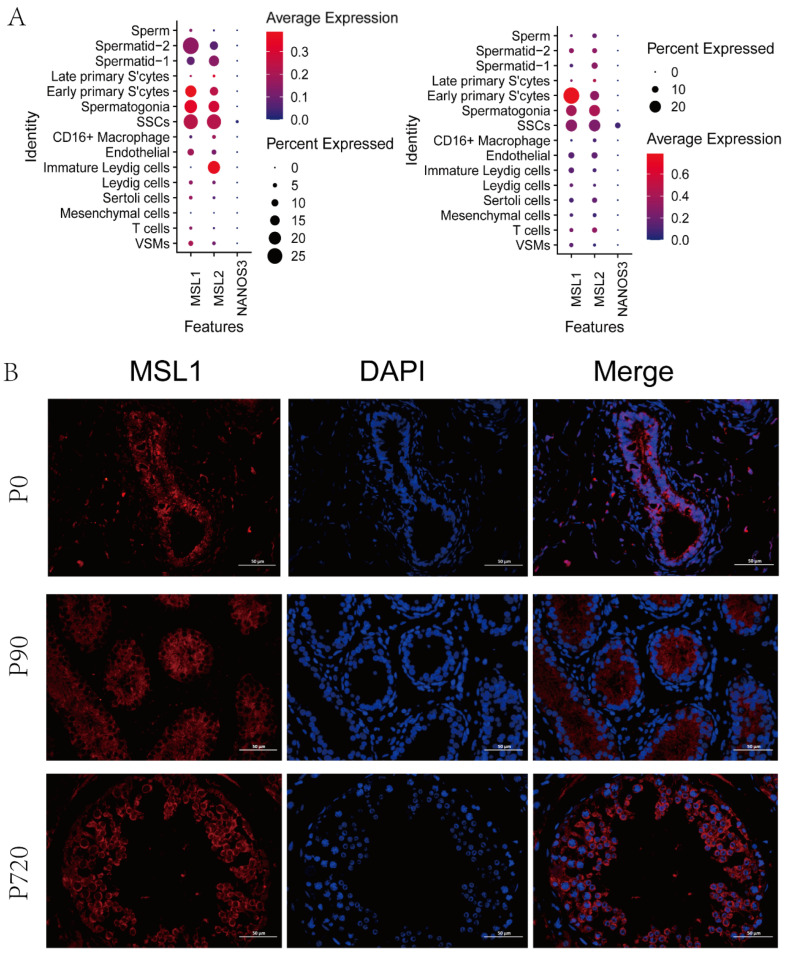
Expression of MSL1 in sheep testis. (**A**). Expression patterns for MSL in testes scRNA-seq data. (**B**). MSL1 immunostaining of sheep testis sections at each age. Bar = 50 μm. DAPI, bule; MSL1, red.

**Table 1 animals-12-02169-t001:** Mean counts per cell from the X chromosome and autosomes in every cell type.

Cell Type	X Counts	Autosome Counts	X/Autosome Ratio
P0			
CD16+ Macrophage	128.5353	4641.2119	0.0277
Endothelial	130.2266	6034.0977	0.0216
Immature Leydig cells	142.4536	5328.6118	0.0267
Leydig cells	75.9112	2764.3817	0.0275
Mesenchymal	70.1564	2510.0474	0.0280
T cells	75.7143	2641.5000	0.0287
VSMs	110.2424	3778.6869	0.0292
Sertoli cells	114.5922	4240.6668	0.0270
SSC	108.9451	5488.4945	0.0198
Spermatogonia	34.9231	1683.3077	0.0207
P30			
CD16+ Macrophage	82.3359	2801.8828	0.0294
Endothelial	132.4890	5846.1923	0.0227
Immature Leydig cells	101.3784	3810.3851	0.0266
Leydig cells	57.7191	2056.5287	0.0281
Mesenchymal	88.0381	3548.5905	0.0248
T cells	43.5422	1807.6024	0.0241
VSMs	85.2736	3195.1857	0.0267
Sertoli cells	124.1719	4505.4355	0.0276
SSC	157.9563	8479.7961	0.0186
Spermatogonia	107.1697	5025.1150	0.0213
Early primary S’cytes	36.0818	5056.9308	0.0071
Late primary S’cytes	5.5809	594.5732	0.0094
Spermatid-1	31.2000	4163.8228	0.0075
Spermatid-2	20.1000	1843.7667	0.0109
Sperm	34.9865	1622.8475	0.0216
P90			
CD16+ Macrophage	62.8087	2341.6242	0.0268
Endothelial	90.9568	4067.5760	0.0224
Immature Leydig cells	107.8000	3975.8000	0.0271
Leydig cells	54.3794	2121.6286	0.0256
Mesenchymal cells	17.0870	791.7826	0.0216
T cells	34.3602	1462.0158	0.0235
VSMs	58.2791	2217.8428	0.0263
Sertoli cells	56.2993	2160.6361	0.0261
SSC	140.0526	7333.1184	0.0191
Spermatogonia	92.8073	4542.2374	0.0204
Early primary S’cytes	37.2829	5132.5854	0.0073
Late primary S’cytes	4.0830	531.4857	0.0077
Spermatid-1	66.8285	10,416.3366	0.0064
Spermatid-2	113.7132	16,945.3000	0.0067
Sperm	14.2388	2233.5853	0.0064

**Table 2 animals-12-02169-t002:** X and autosomal count ratios detected per gene for every cell type.

Cell Type	X/Autosome Ratio of Raw Counts	RAW *p* Value	Adjusted *p* Value
P0			
CD16+ Macrophage	0.58	7.22 × 10^−46^	5.1 × 10^−45^
Endothelial	0.57	9.69 × 10^−44^	5.8 × 10^−43^
Immature Leydig cells	0.55	2.23 × 10^−79^	1.8 × 10-78
Leydig cells	0.54	0	0
Mesenchymal cells	0.54	2.27 × 10^−36^	1.1 × 10^−35^
T cells	0.53	0.00012207	0.00024
VSMs	0.43	5.78 × 10^−18^	2.3 × 10^−17^
Sertoli cells	0.55	0	0
SSCs	0.39	1.21 × 10^−16^	3.6 × 10^−16^
Spermatogonia	0.41	0.000244141	0.00024
P30			
CD16+ Macrophage	0.53	9.69 × 10^−44^	7.8 × 10^−43^
Endothelial	0.48	6.88 × 10^−232^	9.6 × 10^−231^
Immature Leydig cells	0.49	4.95 × 10^−26^	2 × 10^−25^
Leydig cells	0.55	0	0
Mesenchymal cells	0.56	5.92 × 10^−19^	1.8 × 10^−18^
T cells	0.53	2.54 × 10^−15^	5.1 × 10^−15^
VSMs	0.45	4.37 × 10^−52^	3.9 × 10^−51^
Sertoli cells	0.58	1.39 × 10^−85^	1.4 × 10^−84^
SSCs	0.37	1.50 × 10^−35^	9 × 10^−35^
Spermatogonia	0.42	2.62 × 10^−145^	3.4 × 10^−144^
Early primary S’cytes	0.14	7.70 × 10^−28^	3.9 × 10^−27^
Late primary S’cytes	0.19	6.02 × 10^−108^	7.2 × 10^−107^
Spermatid-1	0.15	4.72 × 10^−95^	5.2 × 10^−94^
Spermatid-2	0.22	1.67 × 10^−11^	1.7 × 10^−11^
Sperm	0.43	2.46 × 10^−38^	1.7 × 10^−37^
P720			
CD16+ Macrophage	0.52	1.29863 × 10^−50^	6.5 × 10^−50^
Endothelial	0.47	5.74 × 10^−191^	7.5 × 10^−190^
Immature Leydig cells	0.43	0.0625	0.062
Leydig cells	0.52	0	0
Mesenchymal cells	0.51	2.38419 × 10^−07^	0.00000048
T cells	0.54	2.4637 × 10^−105^	2.7 × 10^−104^
VSMs	0.44	3.1914 × 10^−62^	2.9 × 10^−61^
Sertoli cells	0.53	5.85977 × 10^−50^	2.3 × 10^−49^
SSCs	0.38	3.67866 × 10^−14^	1.1 × 10^−13^
Spermatogonia	0.41	2.00458 × 10^−60^	1.6 × 10^−59^
Early primary S’cytes	0.14	2.77632 × 10^−60^	1.9 × 10^−59^
Late primary S’cytes	0.15	4.5374 × 10^−224^	6.4 × 10^−223^
Spermatid-1	0.13	2.0614 × 10^−52^	1.2 × 10^−51^
Spermatid-2	0.13	1.6026 × 10^−88^	1.6 × 10^−87^
Sperm	0.13	2.199 × 10^−126^	2.6 × 10^−125^

## Data Availability

The data used to support the findings of this study are available from the corresponding author upon request.

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
