# Peer review of "Dosage Compensation of the X Chromosome during Sheep Testis Development Revealed by Single-Cell RNA Sequencing"

_animals, 2022, doi:10.3390/ani12172169_

Round 1

Reviewer 1 Report

I am reviewing the manuscript entitled 'Dosage Compensation of the X Chromosome During Sheep Testis Development Revealed by Single Cell Rna-Sequencing' for consideration to be published in MDPI Animals. 

The manuscript investigates dose compensation of X chromosome in sheep testis development in day 0 90 and 720d. No justification was given for the specific ages. 

The authors used single cell RNA seq to evaluate the gene expression and study the X dose compensation. Results do not support any evidence of dose compensation. 

There are some major points to be improved on this manuscript:

1 - In several sections the writing is confusing. Sentences are long and have many acronyms and parentheses. Additionally, some sentences are confusing and seem to be missing a verb. I highly recommend a carefully review of the entire document. 

2 - The material and methods section, especially the last few sections, seem to be rushed and are not specific enough. Please, add as many details as possible. The last 3 sections must be rewritten. 

3 - The authors do not consider any individual variation of gene expression. This reviewer understands the limitations of large experiments with single cell rna-seq, but n=1 and samples taken at least 90 days apart should at least be acknowledge in the manuscript as a limiting factor of the present study. 

4 - Discussion and introduction must be improved. Better justification of MSL and NANOS3 genes focus should be given. Authors repeat that dose compensation is germ cells is controversial but little to no literature is provided on that. I would expect that at least one paragraph of discussion would be dedicated for such controversy. 

Specific comments 

Simple summary:

Line 19: Please, include why the existence of dose compensation is controversial

Lines 24 and 25: This sentence seems to be out of place. I suggest changing it to a conclusion or something that can be linked to the rest of the summary.

Abstract

Line 28: This sentence could be improved. The abstract starts with a generic definition of dose compensation, and the sentence abruptly shifts to a very specific sheep testis example. 

Lines 32 to 40: The second part of the abstract is confusing. Please, rewrite the section. 

Introduction

Line 60: Please expand the use of reference number 10.

Line 65: Please, review this sentence. It is very long and hard to follow, and there may need grammar review

Line 68: This sentence is hard to follow. Please, consider rewriting it

Line 71: An abrupt change from drosophila and humans to sheep. This need must be justified. 

Material and Methods

Line 68: Sheep instead of sheeps

Line 111: how was this imputation performed? What is the version of R and the package used?

Line 117: Please rewrite this sentence

Lines 116 to 125: Section 2.6 reads more as a result section than material and methods. Consider moving it or rewriting it

Lines 126 to 130: This section must be reviewed.

Lines 132 to 138: Another section that is confusing. It does not read like a scientific paper 

Results 

L 141: Earlier in the paper it stated that the sheep was 0 day instead of 2

L 147: this sentence must be rewritten. 

L 153 to 159: Another sentence that seems to be too long, includes too many acronyms and parenthesis. Please, simplify your format to allow the reader to understand your results. 

Figure 1C: Why were those specific 15 genes selected?

Figure 3: Please, identify the samples in the figure.

L 196: How was difference significance assessed? 

L 208 to 226: Please, justify the special interest in MSL and NANOS3

Discussion 

L256: Please, include citation and discuss the "several species" result. Also, justify why it remains highly controversial

L 266: Please, improve your discussion regarding MSLs 

Reviewer 2 Report

In the present manuscript, Jie Su and colleagues describe the extent of compensated gene expression during testis development in sheep. It is assumed that compensation between X and autosomal gene expression in mammals is achieved by inactivation of one of the two X chromosomes in females and hyperactivation of the remaining X or the single X in males. Although the exact mechanism of this hyperactivation in mammals is not known, variations in this part of the compensation process during gametogenesis can be observed in a variety of animals, including fruit flies.

Here, the authors apply single-cell RNA sequencing to 3 stages of testis development. Because they can distinguish the different cell types, they are able to compare X- and autosomal expression between somatic cells and gametes and conclude that premeiotic germ cells do not show compensated expression.

This descriptive work contains some interesting information for specialists, but for my taste it is not suitable to appeal to a general audience. On the one hand, I attribute this to the relatively small amount of information gained from this very small data set. On the other hand, the analyses performed are not of particularly high quality, and accordingly the results and their presentation lack significance. In addition, there are some minor issues that, along with the major ones, prevent me from recommending this work for publication. 

However, I think that a thorough revision could result in a much better manuscript.

The major problems are: 

1) The genes included in the analysis should be the same for all tissues and time points. Compensated expression cannot be assessed if the tissue and sex specific genes are not removed. The methods do not describe the filtering, so I assume that this selection of genes was not done.

2) The description of the methods is very vague. Each processing step must be documented with software version and parameters. In the data analysis, the number of instances (genes) included in the analysis must be specified. The annotation and genome version must be provided. I strongly recommend depositing the raw data in a public repository (GEO/SRA). At least the counting matrices should be released.

3) The figures are quite difficult to read. Especially the x-scales (gene number, RNA content) should be the same for all panels to allow visual comparison. I would prefer a representation that combines all 3 stages in one panel rather than splitting them. 

Minor issues:

4) In addition the the detailed description of the process in Drosophila melanogaster, a paragraph on mammalian dosage compensation is needed in the introduction. 

5) Statistics in Table 2 are inflated by a large N (number of cells). Robustness cannot be estimated by within-experiment tests. It would be better to omit the tests altogether.

6) The discussion of MSL gene expression makes no sense to me, as their role in dosage compensation in mammals is unclear.

Round 2

Reviewer 1 Report

Thank you for addressing the previous comments. There are still a few points that could be improved, especially regarding the simple summary. There were a few spell and grammar errors that I've found. Please, double check the document once again.

Please, make sure that you have a high-resolution image included in the submission.

Specific comments. 

L 25: However 

L 25 to 27: Please, rewrite those sentences. Citing studies this way in abstract/summary is a bit odd to this reviewer.

L 134: Please define your quality control and low quality reads. 

L 151: Rephrase the sentence: Meanwhile, remove testis-specific or testis-biased genes from data

L 210: please, increase quality of figure2 and 4

L 276: X-chromosome is or X-chromosomes are 

Reviewer 2 Report

2) Response: We thank the reviewer for his/her suggestion.We have revised this section(Materials and Methods section, 2.5 and 2.6, line122-131,line 134-146).We are loading original data to NCBI.

The GEO accession number has to be provided in the manuscript

3) Response: Thanks for pointing this out.We have revised it according to reviewer suggestion.We have combined all 3 stages in one panel(Figure 2, line216).

Figures in the revised manuscript are unchanged. Please revise.

5) Response: Thank you for your comment.Dose compensation exists if individual genes from the X and autosomal chromosomes produce roughly equal amounts of RNA within the same cell type.The number of cells doesn't matter because it's in the same cell type. It is impossible to compare different cell types because of the number of cells.We then log transformed these counts with y = Log2(counts+1) and performed nondirectional Wilcoxon tests, with Holm-corrected p values indicating if genes from the X chromosome are likely to have equal median counts to genes from the autosomes.

In Methods: "dividing this by nGene" is unclear: what is nGene?

Given the null hypothesis of the wilcoxon test being equality of medians and the obvious inequality of medians in all cell types, I strongly suggest to eliminate the statistical tests as they are not providing additional information. 
